# Fractional order *SEIQRD* epidemic model of Covid-19: A case study of Italy

**Subrata Paul[1], Animesh Mahata[2]\*, Supriya Mukherjee[3], Prakash Chandra Mali[4], Banamali Roy[5]**

**1** Department of Mathematics, Arambagh Government Polytechnic, Arambagh, West Bengal, India,
**2** Mahadevnagar High School, Maheshtala, Kolkata, West Bengal, India, **3** Department of Mathematics, Gurudas College, Narkeldanga, Kolkata, West Bengal, India, **4** Department of Mathematics, Jadavpur University, Kolkata, India, **5** Department of Mathematics, Bangabasi Evening College, Kolkata, West Bengal, India

\* animeshmahata8@gmail.com

**Data Availability Statement:** Data may be accessed by any researcher at https://www.worldometers.info/coronavirus/.

## Abstract

The fractional order *SEIQRD* compartmental model of COVID-19 is explored in this manuscript with six different categories in the Caputo approach. A few findings for the new model's existence and uniqueness criterion, as well as non-negativity and boundedness of the solution, have been established. When $R_{Covid19} < 1$ at infection-free equilibrium, we prove that the system is locally asymptotically stable. We also observed that $R_{Covid\ 19} < 1$, the system is globally asymptotically stable in the absence of disease. The main objective of this study is to investigate the COVID-19 transmission dynamics in Italy, in which the first case of Coronavirus infection 2019 (COVID-19) was identified on January 31st in 2020. We used the fractional order *SEIQRD* compartmental model in a fractional order framework to account for the uncertainty caused by the lack of information regarding the Coronavirus (COVID-19). The Routh-Hurwitz consistency criteria and La-Salle invariant principle are used to analyze the dynamics of the equilibrium. In addition, the fractional-order Taylor's approach is utilized to approximate the solution to the proposed model. The model's validity is demonstrated by comparing real-world data with simulation outcomes. This study considered the consequences of wearing face masks, and it was discovered that consistent use of face masks can help reduce the propagation of the COVID-19 disease.

## 1. Introduction

The world is still addressing the Coronavirus illness 2019 (COVID-19), which is caused by the new Coronavirus SARSCoV-2, a highly aggressive virus that attacks the individual respiratory system. The hospitalized individuals' ailments were linked to the marine and moist animal industries in Wuhan, Hubei Province, China [1]. COVID-19 spreads from person to person by touching contaminated surfaces and inhalation of infected persons' respiratory droplets [2]. Those who have been infected with COVID-19 have reported high fever, persistent cough, and exhaustion. Nonetheless, depending on the immune system, COVID-19 symptoms and consequences differ from person to person. People with a strong immune response seem to be more

**Funding:** The author(s) received no specific funding for this work.

**Competing interests:** The authors have declared that no competing interests exist.

likely to get mild—to—moderate illnesses as well as recover avoid going to the hospital. Various investigations, however, have identified other symptoms such as neurological illnesses and gastroenteritis of different severity [3, 4]. With so many waves of infection, the illness caused numerous deaths in many countries. COVID-19 outbreaks have occurred in Italy, with the population suffering the effects of the consequences. The number of confirmed incidence and mortality in every phase has been published, and there appears to be an increasing incidence. On February 21, 2020, the first Italian victim of COVID-19, a 38-year-old male hospitalized at Codogno Hospital in Lodi, was diagnosed. On the 12[th] of February, 2022, it has infected over 424,636,034 people over the world, resulting in 5903,485 deaths and 349,857,774 recoveries [5]. According to reports, the mortality rate in waves 1 and 2 was 1%. Many social programmers and events have been discontinued or extended as a result of the epidemic. The T-20 cricket world cup will be hosted in Australia in 2020, while the Summer Olympics, which were scheduled to be held in Tokyo, have been postponed. The Indian Premier League, one of the most popular cricket events, has been relocated from India to the United Arab Emirates.

Its importance has been demonstrated by the construction of mathematical models in the fields of epidemiology and physics. The Coronavirus infection has been examined by several researchers from various perspectives. While biologists and mathematicians working on the systems of the COVID-19 disease analyzed and constructed mathematical systems based on real-world cases from various countries, and offered information on the infection's peak and clearance. In this context [4, 5], are some mathematical models that have been developed for this disease. The information from Italy is taken into account, and a mathematical model for the COVID-19 disease is developed, with its study reported in [6, 7]. Examines the number of genuine instances from the Mexican population using a mathematical model [8]. Proposes a fractional SEIR model utilizing the wavelet approach. The authors investigated the influence of social distance and other factors that might be regarded important for the reduction of COVID-19 infection in [9]. The authors used a mathematical modeling technique to evaluate genuine infected patients from Saudi Arabia and generated results on disease eradication in the nation [10, 11], Describes a comparative study of Coronavirus infection dynamics. In [12, 13] suggests some additional relevant work on COVID-19 modeling and associated illness outcomes. In [14], Paul et al. analyzed the scenario analysis of COVID-19 pandemic using SEIR epidemic model. A deeper understanding of the pandemic dynamics, including the characteristics of Covid-19 transmission, was made possible by the modeling technique [15, 16]. We have previously published research on fractional order phenomena [17–19]. The novel fractional operator has shown to be quite effective in solving a variety of mathematical modeling problems as well as some recent work on COVID-19 [20–23]. From the study of data obtained from Wuhan, Li et al. [24] calculated the epidemiology and discovered the mean incubation time was 5.2 days. The authors of [25, 26] highlighted some interesting outcomes of Corona virus disease. In [27] introduced deeper investigation of modified epidemiological computer virus model containing the Caputo operator. Furthermore, the researchers in [28–34] analyzed fractional derivatives of the COVID-19 infection models and presented some recommendations for infection minimization in the form of lockdown and control measures.

## 1.1 Motivation and research background

Fractional order modeling is a useful tool that has been used to explore the nature of diseases since the fractional derivative is an extension of the integer-order derivative. In order to replicate real-world issues, several innovative fractional operators with various properties have been designed. In addition, the integer derivative has a local identity, whereas the fractional derivative has a global character. Numerous varieties of fractional derivatives, both with and

without singular kernels, are available today. Leibniz's query from 1695 marks the beginning of the fractional derivative. The fractional derivative also improves in the improvement of the system's consistency domain. We have the derivatives of Caputo, Riemann-Liouville, and Katugampola for singular kernels [35, 36]. There are two varieties of fractional derivatives without singular kernels: the Caputo-Fabrizio fractional derivative [37], which has an exponential kernel, and the Atangana-Baleanu fractional derivative, which has a Mittag-Leffler kernel [38]. While memory and genetic properties are involved, working with fractional-order derivatives is crucial because it provides a more accurate technique to describe COVID-19 outbreaks. Numerous academic articles, monographs, and novels have provided evidence to support this claim; for instance, [39–46]. Motivated by the current research, we present and analyze the *SEIQRD* model in Caputo sense. The Caputo derivative is particularly useful for discussing real-world situations since it permits traditional beginning and boundary conditions to be used in the derivation, and the derivative of a constant is zero, whereas the Riemann–Liouville fractional derivative does not. It is quite challenging to genuinely create an appropriate mathematical model using classical differentiation in the situation of COVID-19 because to the large number of uncertainties, unknowns, and disinformation. Generally, non-local operators are better suited for such circumstances because, depending on whether power law, fading memory, or overlap effects are taken into account, they can represent non-localities and certain memory effects.

## 1.2 Structure of the paper

We present the reader with some important definitions and characteristics of fractional derivatives in Section 2. In Section 3, we have established the *SEIQRD* epidemic model of Covid-19 in Caputo sense. We have investigated the existence, uniqueness, non-negative, boundedness criterion and stability analysis of the solution of model in Section 4. In Section 5, the fractional-order Taylor's approach in Caputo derivative is utilized to approximate the solution to the proposed model. The numerical study is given using MATLAB (2018a) in Section 6. Finally, the paper's conclusion is found in Section 7.

## 2. Preliminaries

We provide the reader with some useful definitions and characteristics of fractional derivatives.

**Definition 1** [37] "The Caputo fractional derivative of order $0<\phi\leq1$ for the function $u$: $C^n[0, \infty]\rightarrow\mathbb{R}$ is defined as

$$^CD_t^\phi\left(u(t)\right) = \frac{1}{\Gamma(n-\phi)}\int_0^t\frac{1}{(t-z)^{\phi+1-n}}\frac{d^n}{dz^n}u(z)dz,$$

where $C^n[0, \infty]$ is a $n$ tines continuously differentiable function and the Gamma function is defined by $\Gamma()$ such that $n-1<\phi<n$".

**Theorem 1** [47] "If $^CD_t^\phi\ u(t)$ is piecewise continuous, then $L(^CD_t^\phi\left(u(t)\right) = z^\phi L(u(t)) - \sum_{i=0}^{l-1}z^{\phi-i-1}u^{(i)}(0), l-1 < \phi \leq l \in \mathbb{N}$, where the Laplace transform is denoted by $L(g(t))$".

**Theorem 2** [48] "One-parametric and two-parametric Mittag-Leffler functions are described as follows: $E_{a_1}(z) = \sum_{i=0}^\infty\frac{z^i}{\Gamma(a_1i+1)}$ and $E_{a_1,a_2}(z) = \sum_{i=0}^\infty\frac{z^i}{\Gamma(a_1i+a_2)}$, where $a_1, a_2\in\mathbb{R}^+$".

**Lemma 1** [49] "Let $0<\phi\leq1$, $u(t)\in C[p, q]$ and if $^CD_t^\phi\ u(t)$ is continuous in $[p, q]$, then $u(x) = u(p) + \frac{1}{\Gamma(\phi)}(x-p)^\phi.\ ^CD_t^\phi\ u(z)$,

where $0\leq z\leq x, \forall x\in(p, q]$".

**Note 1** "If $^{C}D_t^{\phi}\, u(t) \geq 0 (^{C}D_t^{\phi}\, u(t) \leq 0), t \in (p,q)$, then $u(t)$ is a non-decreasing (non-increasing) function for $t \in [p,q]$".

**Lemma 2** "Let us consider the fractional order system as

$$^{C}D_t^{\phi}\, (Y(t)) = \Psi(Y), Y_{t_0} = (y_{t_0}{}^1, y_{t_0}{}^2, \ldots, y_{t_0}{}^n), y_{t_0}{}^j, j = 1, 2, \ldots, n,$$

with $0 < \phi < 1$, $Y(t) = (y^1(t), y^2(t), \ldots, y^n(t))$ and $\Psi(Y) : [t_0, \infty] \to \mathbb{R}^{n \times n}$. For calculate the equilibrium points, we have $\Psi(Y) = 0$. These equilibrium points are locally asymptotically stable iff each eigen value $\lambda_j$ of the Jacobian matrix $J(Y) = \frac{\partial(\Psi_1, \Psi_2, \ldots, \Psi_n)}{\partial(y^1, y^2, \ldots, y^n)}$ calculated at the equilibrium points satisfies $|\arg(\lambda_j)| > \frac{\phi\pi}{2}$".

**Lemma 3** "Assume that $u(t) \in \mathbb{R}^+$ is a differentiable function. Then, for any $t > 0$,

$$^{C}D_t^{\phi}\left[u(t) - u^* - u^* ln\frac{u(t)}{u^*}\right] \leq \left(1 - \frac{u^*}{u(t)}\right)\, ^{C}D_t^{\phi}\, (u(t)), u^* \in \mathbb{R}^+, \forall \phi \in (0,1)".$$

## 3. Model formulation

The mathematical model of COVID-19 transmission formulated in this study was motivated by the study of [14, 17, 18]. In the present study, the model will be divided into six compartments [see **Fig 1**]. The total human population to be considered is denoted as $N(t)$, and at any

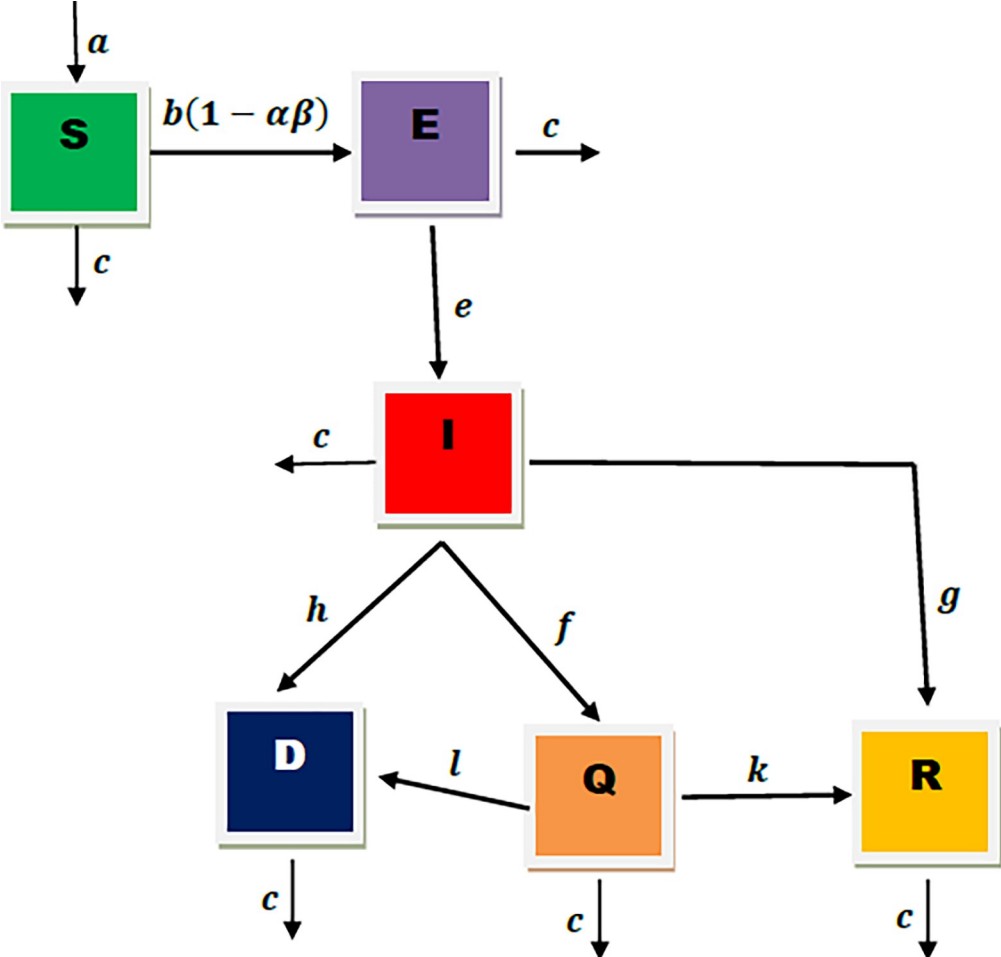

**Fig 1. Depicts a flow chart of the proposed *SEIQRD* model.**

time, it comprises of the susceptible ($S$), exposed ($E$), infected ($I$), quarantined ($Q$), recovered ($R$), and death ($D$) compartments, respectively.

$$\text{Thus } N(t) = S(t) + E(t) + I(t) + Q(t) + R(t) + D(t). \tag{3.1}$$

Now we formulate the SEIQRD model with fractional order derivatives with Caputo operator of order $0 < \phi \leq 1$.

$$^{C}D_t^{\phi}\, S(t) = \breve{a}^{\phi} - \breve{b}^{\phi}(1 - \breve{\alpha}^{\phi}\breve{\beta}^{\phi})S(t)I(t) - \breve{c}^{\phi}S(t),$$

$$^{C}D_t^{\phi}\, E(t) = \breve{b}^{\phi}(1 - \breve{\alpha}^{\phi}\breve{\beta}^{\phi})S(t)I(t) - (\breve{e}^{\phi} + \breve{c}^{\phi})E(t), \tag{3.2}$$

$$^{C}D_t^{\phi}\, I(t) = \breve{e}^{\phi}E(t) - (\breve{f}^{\phi} + \breve{g}^{\phi} + \breve{h}^{\phi} + \breve{c}^{\phi})I(t),$$

$$^{C}D_t^{\phi}\, Q(t) = \breve{f}^{\phi}I(t) - (\breve{k}^{\phi} + \breve{l}^{\phi} + \breve{c}^{\phi})Q(t),$$

$$^{C}D_t^{\phi}\, R(t) = \breve{g}^{\phi}I(t) + \breve{k}^{\phi}Q(t) - \breve{c}^{\phi}R(t),$$

$$^{C}D_t^{\phi}\, \mathrm{D}(\mathrm{t}) = \breve{h}^{\phi}I(t) + \breve{l}^{\phi}Q(t).$$

Now for the sake of convenience of calculation, we redefine the parameters [see **Table 1**] as $a = \breve{a}^{\phi}, b = \breve{b}^{\phi}, \alpha = \breve{\alpha}^{\phi}, \beta = \breve{\beta}^{\phi}, c = \breve{c}^{\phi}, e = \breve{e}^{\phi}, f = \breve{f}^{\phi}, g = \breve{g}^{\phi}, h = \breve{h}^{\phi}, l = \breve{l}^{\phi}, k = \breve{k}^{\phi}$. Thus, the modified model system (3.2) can be finally written in the following form with initial conditions,

$$^{C}D_t^{\phi}\, S(t) = a - b(1 - \alpha\beta)S(t)I(t) - cS(t),$$

$$^{C}D_t^{\phi}\, E(t) = b(1 - \alpha\beta)S(t)I(t) - (e + c)E(t), \tag{3.3}$$

$$^{C}D_t^{\phi}\, I(t) = eE(t) - (f + g + h + c)I(t),$$

$$^{C}D_t^{\phi}\, Q(t) = fI(t) - (k + l + c)Q(t),$$

**Table 1. Description of the relevant parameters.**

| Parameters | Significance |
| --- | --- |
| $a$ | Recruitment rate into $S$ |
| $b$ | Contact rate |
| $\alpha$ | Percentage of people who use a face mask |
| $\beta$ | The efficacy of face masks |
| $c$ | Mortality rate of all individuals |
| $e$ | Progression rate from $E$ to $I$ |
| $f$ | Isolation rate for $I$ |
| $g$ | Recovery rate of $I$ |
| $h$ | Death rate of $I$ due to COVID-19 disease |
| $k$ | Recovery rate of $Q$ |
| $l$ | Death rate of $Q$ due to COVID-19 disease |

$${}^{C}D_t^{\phi} R(t) = gI(t) + kQ(t) - cR(t),$$

$${}^{C}D_t^{\phi} D(t) = hI(t) + lQ(t).$$

The initial conditions are

$$S(0) > 0, E(0) > 0, I(0) > 0, Q(0) > 0, R(0) > 0, D(0) > 0. \tag{3.4}$$

## 4. Analysis of the system

**4.1 Existence and uniqueness.** The following are the necessary and sufficient conditions for a fractional order system's solution to exist and be unique:

**Theorem 4.1.1.** For each initial condition, there exists a unique solution of fractional order system (3.3).

Proof We are looking for a sufficient condition for the presence and uniqueness of system (3.3) solutions in the region $\Pi \times (0, T]$ where

$\Pi = \{(S, E, I, Q, R, D) \in \mathbb{R}^6 : max\|S\|, \|E\|, \|I\|, \|Q\|, \|R\|, \|D\| \leq M\}$. The method employed in [33] is used. Consider a mapping $F(Y) = (F_1(Y), F_2(Y), F_3(Y), F_4(Y), F_5(Y), F_6(Y))$ where $Y = (S, E, I, Q, R, D)$ and $\bar{Y} = (\bar{S}, \bar{E}, \bar{I}, \bar{Q}, \bar{R}, \bar{D})$:

$$F_1(Y) = a - b(1 - \alpha\beta)S(t)I(t) - cS(t),$$

$$F_2(Y) = b(1 - \alpha\beta)S(t)I(t) - (e + c)E(t),$$

$$F_3(Y) = eE(t) - (f + g + h + c)I(t),$$

$$F_4(Y) = fI(t) - (k + l + c)Q(t),$$

$$F_5(Y) = gI(t) + kQ(t) - cR(t),$$

$$F_6(Y) = hI(t) + lQ(t).$$

For any $Y, \bar{Y} \in \Pi$ :

$$\begin{aligned}
&\|F(Y) - F(\bar{Y})\| \\
&= |F_1(Y) - F_1(\bar{Y})| + |F_2(Y) - F_2(\bar{Y})| + |F_3(Y) - F_3(\bar{Y})| + |F_4(Y) - F_4(\bar{Y})| + |F_5(Y) \\
&\quad - F_5(\bar{Y})| + |F_6(Y) - F_6(\bar{Y})|
\end{aligned}$$

$$\begin{aligned}
&= |a - b(1 - \alpha\beta)S(t)I(t) - cS(t) - a + b(1 - \alpha\beta)\bar{S}(t)\bar{I}(t) + c\bar{S}(t)| + |b(1 - \alpha\beta)S(t)I(t) - (e \\
&\quad + c)E(t) - b(1 - \alpha\beta)\bar{S}(t)\bar{I}(t) + (e + c)\bar{E}(t)| + |eE(t) - (f + g + h + c)I(t) - e\bar{E}(t) + (f \\
&\quad + g + h + c)\bar{I}(t)| + |fI(t) - (k + l + c)Q(t) - f\bar{I}(t) + (k + l + c)\bar{Q}(t)| + |gI(t) + kQ(t) \\
&\quad - cR(t) - g\bar{I}(t) - k\bar{Q}(t) + c\bar{R}(t)| + |hI(t) + lQ(t) - h\bar{I}(t) - l\bar{Q}(t)|
\end{aligned}$$

$$= |b(1 - \alpha\beta)(\bar{S}(t)\bar{I}(t) - S(t)I(t)) + c(\bar{S}(t) - S(t))| + |b(1 - \alpha\beta)(S(t)I(t) - \bar{S}(t)\bar{I}(t)) - (e$$
$$+ c)(E(t) - \bar{E}(t))| + |e(E(t) - \bar{E}(t)) - (f + g + h + c)(I(t) - \bar{I}(t))| + |f(I(t) - \bar{I}(t)) - (k$$
$$+ l + c)(Q(t) - \bar{Q}(t))| + |g(I(t) - \bar{I}(t)) + k(Q(t) - \bar{Q}(t)) - c(R(t) - \bar{R}(t))| + |h(I(t)$$
$$- \bar{I}(t)) + l(Q(t) - \bar{Q}(t))|$$

$$\leq (c + 2b(1 - \alpha\beta)M)|(\bar{S}(t) - S(t))| + (2e + c)|(E(t) - \bar{E}(t))| + (2f + 2g + 2h + c)|(I(t)$$
$$- \bar{I}(t))) + (2k + 2l + c)|(Q(t) - \bar{Q}(t))) + c|(R(t) - \bar{R}(t))|$$

$$\leq G_1|S - \bar{S}| + G_2|E - \bar{E}| + G_3|I - \bar{I}| + G_4|Q - \bar{Q}| + G_5|R - \bar{R}| \leq G\|Y - \bar{Y}\|.$$

Where $G = \max\{G_1, G_2, G_3, G_4, G_5\}$ and $G_1 = (c + 2b(1 - \alpha\beta)M), G_2 = (2e + c), G_3 = (2f + 2g + 2h + c), G_4 = (2k + 2l + c), G_5 = c$.

As a result, F(Y) fulfils the Lipschitz requirement. As a consequence, fractional order system (3.3) exists and is unique.

## 4.2 Non-negativity and boundedness of proposed model

**Proposition** The region $\Omega = \left\{(S, E, I, Q, R, D) \in \mathbb{R}^6 : 0 < N \leq \frac{a}{c}\right\}$ is non-negative invariant for the model (3.3) $\forall\, t \geq 0$.

Proof We have

$$^C D_t^\phi (S + E + I + Q + R + D)(t) = a - c(S + E + I + Q + R + D)(t)$$

$$\Rightarrow\, ^C D_t^\phi N(t) = a - cN(t)$$

$$\Rightarrow\, ^C D_t^\phi N(t) + cN(t) = a. \tag{4.1}$$

Using Laplace transform and Theorem 7.2 in [50], we have
$z^\phi L(N(t)) - z^{\phi-1}N(0) + cL(N(t)) = \frac{a}{z}$, where z is the Laplace transform parameter.

$$\Rightarrow\, L(N(t))(z^{\phi+1} + c) = z^\phi N(0) + a$$

$$\Rightarrow\, L(N(t)) = \frac{z^\phi N(0) + a}{z^{\phi+1} + c} = \frac{z^\phi N(0)}{z^{\phi+1} + c} + \frac{a}{z^{\phi+1} + c}. \tag{4.2}$$

Appling inverse Laplace transform, we have

$$N(t) = N(0)E_{\phi,1}(-ct^\phi) + at^\phi E_{\phi,\phi+1}(-ct^\phi).$$

According to Mittag-Leffler function,

$$E_{c,d}(x) = x\, E_{c,c+d}(x) + \frac{1}{\Gamma(d)}.$$

Hence, $N(t) = \left(N(0) - \frac{a}{c}\right)E_{\phi,1}(-ct^\phi) + \frac{a}{c}$.

$$\text{Thus } \lim_{t \to \infty} Sup\, N(t) \leq \frac{a}{c}. \tag{4.3}$$

As a result, the functions S, E, I, Q, R, and D are all non-negative.

## 4.3 The equilibrium points of the system

The system's equilibrium may be found by solving the model (3.3) i.e.,

$$^{C}D_t^\phi\, S(t) = {^{C}D_t^\phi}\, E(t) = {^{C}D_t^\phi}\, I(t) = {^{C}D_t^\phi}\, Q(t) = {^{C}D_t^\phi}\, R(t) = {^{C}D_t^\phi}\, D(t) = 0. \tag{4.4}$$

The model (3.3) has two equilibrium points namely, the infection free equilibrium $E_0 = \left(\frac{a}{c}, 0, 0, 0, 0, 0\right)$ and the epidemic equilibrium point $E_1 = (S^*, E^*, I^*, Q^*, R^*, D^*)$, where $S^* = \frac{(e+c)(f+g+h+c)}{be(1-\alpha\beta)}, E^* = \frac{(f+g+h+c)}{e} I^*, I^* = \frac{a}{b(1-\alpha\beta)S^*} - \frac{c}{b(1-\alpha\beta)} = \frac{ae}{(e+c)(f+g+h+c)} - \frac{c}{b(1-\alpha\beta)} = \frac{c}{b(1-\alpha\beta)}(R_{Covid19} - 1), Q^* = \frac{f}{(k+l+c)} I^*, R^* = \left(g + \frac{kf}{k+l+c}\right) I^*, D^* = 0.$

## 4.4 The basic reproduction number of the system

The next-generation matrix technique is used to calculate the model's basic reproduction number $R_{Covid\ 19}$, which may be obtained from the maximum eigen value of the matrix $\mathcal{F}V^{-1}$ [51, 52] where,

$$\mathcal{F} = \begin{bmatrix} 0 & b(1-\alpha\beta)\frac{a}{c} \\ 0 & 0 \end{bmatrix} \text{ and } V = \begin{bmatrix} e+c & 0 \\ -e & f+g+h+c \end{bmatrix}.$$

Therefore, $R_{Covid19} = \dfrac{bae(1-\alpha\beta)}{c(e+c)(f+g+h+c)}.$ \qquad (4.5)

## 4.5 Stability behavior at $E_0$

The Jacobian matrix of the model (3.3) at $E_0$ is given by $J_{E_0} = A$, where

$$A = \begin{bmatrix} A_{11} & 0 & A_{13} & 0 & 0 & 0 \\ 0 & A_{22} & A_{23} & 0 & 0 & 0 \\ 0 & A_{32} & A_{33} & 0 & 0 & 0 \\ 0 & 0 & A_{43} & A_{44} & 0 & 0 \\ 0 & 0 & A_{53} & A_{54} & A_{55} & 0 \\ 0 & 0 & A_{63} & A_{64} & 0 & 0 \end{bmatrix},$$

with $A_{11} = -a, A_{22} = -(e+c), A = e, A_{13} = -\frac{ba(1-\alpha\beta)}{c}, A_{23} = \frac{ba(1-\alpha\beta)}{c}, A_{33} = -(f+g+h+c),$ $A_{43} = f, A_{53} = g, A_{63} = h, A_{44} = -(k+l+c), A_{54} = k, A_{64} = l, A_{55} = -c.$

**Theorem 4.5.1.** When $R_{Covid\ 19} < 1$, the system (3.3) is globally asymptotically stable, and unstable when $R_{Covid\ 19} > 1$ at $E_0$.

Proof Using the appropriate Lyapunov function

$$\digamma = (e)\mathrm{E} + (e+c)\mathrm{I}.$$

The aforementioned function's time derivative is

$$^{C}D_t^\phi\, \digamma(\mathrm{t}) = (e)^{C}D_t^\phi\, \mathrm{E(t)} + (e+c)^{C}D_t^\phi\, \mathrm{I(t)}.$$

From (3.3) we get,

$$^{C}D_t^\phi\, \digamma(\mathrm{t}) = (e)[b(1-\alpha\beta)SI - (e+c)E] + (e+c)[eE - (f+g+h+c)I].$$

Now,

$$^{C}D_{t}^{\phi} \mathcal{F}(t) = be(1 - \alpha\beta)SI - (e + c)(f + g + h + c)I$$

$$= I[(e + c)(f + g + h + c)]\left[\frac{be(1 - \alpha\beta)S}{(e + c)(f + g + h + c)} - 1\right].$$

Since $S = \frac{a}{c} \leq N$, it follows that

$$^{C}D_{t}^{\phi} \mathcal{F}(t) = I[(e + c)(f + g + h + c)]\left[\frac{abe(1 - \alpha\beta)}{c(e + c)(f + g + h + c)} - 1\right]$$

$$= I[(e + c)(f + g + h + c)][R_{Covid19} - 1].$$

Hence if $R_{Covid\ 19} < 1$, then $^{C}D_{t}^{\phi} \mathcal{F}(t) < 0$.

As a result of LaSalle's use of Lyapunov's concept [53, 54], the point $E_0$ is globally asymptotically stable and unstable if $R_{Covid\ 19} > 1$.

## 4.6 Stability behavior at $E_1$

**Theorem 4.6.1.** If $R_{Covid\ 19} > 1$, the system (3.3) is globally asymptotically stable at $E_1$.

Proof The Lyapunov function of the Goh-Volterra form's is as follows:

$$W = \left(S - S^{*} - S^{*}log\frac{S}{S^{*}}\right) + \left(E - E^{*} - E^{*}log\frac{E}{E^{*}}\right) + L\left(I - I^{*} - I^{*}log\frac{I}{I^{*}}\right).$$

Using Lemma 3 and taking Caputo derivative, we get

$$^{C}D_{t}^{\phi} W(t) \leq \left(1 - \frac{S^{*}}{S}\right){}^{C}D_{t}^{\gamma} S(t) + \left(1 - \frac{E^{*}}{E}\right){}^{C}D_{t}^{\gamma}(t) + L\left(1 - \frac{I^{*}}{I}\right){}^{C}D_{t}^{\gamma} I(t). \qquad (4.6)$$

Using (3.3) we get,

$$^{C}D_{t}^{\phi} W(t) \leq \left(a - b(1 - \alpha\beta)SI - cS - \frac{S^{*}(a - b(1 - \alpha\beta)SI - cS)}{S}\right)$$

$$+ \left((b(1 - \alpha\beta)SI - (e + c)E) - \frac{E^{*}(b(1 - \alpha\beta)SI - (e + c)E)}{E}\right)$$

$$+ L\left((eE - (f + g + h + c)I) - \frac{I^{*}(eE - (f + g + h + c)I)}{I}\right). (4.7)$$

Eq (3.3) gives us the steady state,

$$a = b(1 - \alpha\beta)S^{*}I^{*} + cS^{*}. \qquad (4.8)$$

Substituting Eq (4.8) into (4.7) we have

$$^{C}D_{t}^{\phi} W(t) \leq \left(b(1 - \alpha\beta)S^{*}I^{*} + cS^{*} - b(1 - \alpha\beta)SI - cS - \frac{S^{*}(b(1 - \alpha\beta)S^{*}I^{*} + cS^{*} - b(1 - \alpha\beta)SI - cS)}{S}\right)$$

$$+ \left((b(1 - \alpha\beta)SI - (e + c)E) - \frac{E^{*}(b(1 - \alpha\beta)SI - (e + c)E)}{E}\right)$$

$$+ L\left((eE - (f + g + h + c)I) - \frac{I^{*}(eE - (f + g + h + c)I)}{I}\right).$$

Further simplification gives,

$$
\begin{aligned}
{}^{C}D_t^{\phi}\,W(t) &\leq \left(b(1-\alpha\beta)S^*I^* + cS^* - b(1-\alpha\beta)SI - cS - \frac{S^*(b(1-\alpha\beta)S^*I^* + cS^* - b(1-\alpha\beta)SI - cS)}{S}\right) \\
&\quad + \left((-(e+c)E) - \frac{E^*(b(1-\alpha\beta)SI - (e+c)E)}{E}\right) \\
&\quad + L\left((eE - (f+g+h+c)I) - \frac{I^*(eE - (f+g+h+c)I)}{I}\right).
\end{aligned} \tag{4.9}
$$

Taking all infected classes that do not have a single star (*) from (4.9) and equal to zero:

$$
b(1-\alpha\beta)S^* - (e+c)E + L(eE - (f+g+h+c)I) = 0. \tag{4.10}
$$

The steady state was slightly perturbed between (3.3) and (4.10), resulting in:

$$
L = \frac{b(1-\alpha\beta)S^*}{(f+g+h+c)}, (e+c) = \frac{I^*b(1-\alpha\beta)S^*}{E^*}, e = \frac{(f+g+h+c)I^*}{E^*}. \tag{4.11}
$$

Using (4.11) into (4.9) gives:

$$
\begin{aligned}
{}^{C}D_t^{\phi}\,W(t) &\leq (b(1-\alpha\beta)S^*I^* + cS^* - cS - \frac{S^*(b(1-\alpha\beta)S^*I^* + cS^* - cS)}{S}) + (-\frac{E^*b(1-\alpha\beta)SI}{E} \\
&\quad + I^*b(1-\alpha\beta)S^*) + (-\frac{I^*S^*Eb(1-\alpha\beta)}{IE^*} + b(1-\alpha\beta)S^*I^*).
\end{aligned}
$$

Using $A.\,M \geq G.\,M.$, we have $\left(2 - \frac{S}{S^*} - \frac{S^*}{S}\right) \leq 0$, $\left(3 - \frac{S^*}{S} - \frac{I^*E}{IE^*} - \frac{SE^*I}{E}\right) \leq 0$.

Thus, ${}^{C}D_t^{\phi}\,W(t) \leq 0$.

The point $E_1$ is globally asymptotically stable if $R_{Covid\ 19} > 1$.

## 5. Numerical procedure

As discussed in Theorem 4.1.1, the solution of the system (3.3) is unique. To obtain the numerical solution of the system (3.3), Taylor's theorem will be used.

As a result, we proceed with the model's $1^{st}$ equation as follows:

$$
\begin{cases}
{}^{C}D_t^{\phi}\,S(t) = \Lambda_1(t, S, E, I, Q, R, D), \\
S(0) = S_0, t > 0.
\end{cases} \tag{5.1}
$$

Consider the set of points $[0, A]$ as the points on which we are prepared to approximate the system's solution. Actually, we are unable to calculate S(t), which will be the system's necessary solution. We divide $[0, A]$, into $P$ subintervals $[t_r, t_{r+1}]$ of length, i.e., $m = \frac{A}{P}$, by using the nodes $t_r = rm$, for $r = 0, 1, 2, \ldots, P$. We extend the Taylor's theorem at about $t = t_0$, we have a constant $k \in [0, A]$, such that

$$
S(t) = S(t_0) + {}^{C}D_t^{\phi}\,S(t)\left\{\frac{m^{\phi}}{\Gamma(\phi+1)}\right\} + {}^{C}D_t^{2\phi}[S(t)]_{t=k}\left\{\frac{m^{2\phi}}{\Gamma(2\phi+1)}\right\}. \tag{5.2}
$$

Now substitute $^{C}D_t^{\phi} S(t_0) = \Lambda_1(t_0, S(t_0), E(t_0), I(t_0), Q(t_0), R(t_0), D(t_0))$, and $t = t_1$ in (5.2), which provides

$$S(t_1) = S(t_0) + \Lambda_1(t_0, S(t_0), E(t_0), I(t_0), Q(t_0), R(t_0), D(t_0)) \left\{ \frac{m^{\phi}}{\Gamma(\phi+1)} \right\}$$
$$+ {}^{C}D_t^{2\phi}[S(t)]_{t=k} \left\{ \frac{m^{2\phi}}{\Gamma(2\phi+1)} \right\} \tag{5.3}$$

If $m$ is small, we ignore the higher terms, then (5.3), implies

$$S(t_1) = S(t_0) + \Lambda_1(t_0, S(t_0), E(t_0), I(t_0), Q(t_0), R(t_0), D(t_0)) \left\{ \frac{m^{\phi}}{\Gamma(\phi+1)} \right\}. \tag{5.4}$$

A general formula of expanding about $t_r = t_r + m$, is

$$S(t_{r+1}) = S(t_r) + \Lambda_1(t_r, S(t_r), E(t_r), I(t_r), Q(t_r), R(t_r), D(t_r)) \left\{ \frac{m^{\phi}}{\Gamma(\phi+1)} \right\}. \tag{5.5}$$

In similar way, we get

$$E(t_{r+1}) = E(t_r) + \Lambda_1(t_r, S(t_r), E(t_r), I(t_r), Q(t_r), R(t_r), D(t_r)) \left\{ \frac{m^{\phi}}{\Gamma(\phi+1)} \right\}. \tag{5.6}$$

$$I(t_{r+1}) = I(t_r) + \Lambda_1(t_r, S(t_r), E(t_r), I(t_r), Q(t_r), R(t_r), D(t_r)) \left\{ \frac{m^{\phi}}{\Gamma(\phi+1)} \right\}. \tag{5.7}$$

$$Q(t_{r+1}) = Q(t_r) + \Lambda_1(t_r, S(t_r), E(t_r), I(t_r), Q(t_r), R(t_r), D(t_r)) \left\{ \frac{m^{\phi}}{\Gamma(\phi+1)} \right\}. \tag{5.8}$$

$$R(t_{r+1}) = R(t_r) + \Lambda_1(t_r, S(t_r), E(t_r), I(t_r), Q(t_r), R(t_r), D(t_r)) \left\{ \frac{m^{\phi}}{\Gamma(\phi+1)} \right\}. \tag{5.9}$$

$$D(t_{r+1}) = D(t_r) + \Lambda_1(t_r, S(t_r), E(t_r), I(t_r), Q(t_r), R(t_r), D(t_r)) \left\{ \frac{m^{\phi}}{\Gamma(\phi+1)} \right\}. \tag{5.10}$$

## 6. Numerical study

Numerical simulations employing Taylor's theorem are carried out with the help of MATLAB software to support the mathematical study of the system (3.3). This section is divided into four parts. The stability of our proposed model is discussed at $E_0$ and $E_1$ in Part 1. Part 2 delves into the dynamical behavior of all individuals of various fractional orders. Part 3 is to explore the varying effects of face masks. Part 4 is to determine whether the model (3.3) fits the data. One of the key components in the verification of an epidemiological model is the fitting of the parameters. We ran numerical simulations to contrast the output of our model with actual data from a number of reports released by the WHO and worldometer [5, 6]. Italy has a population of about 60,278,248 people [55]. In Italy, there are 7.2 births per 1000 people [56]. The computed recruiting rate is $\frac{7.21 \times 60278248}{1000 \times 365} = 1191$ per day.

### Part 1

The stability of our suggested model is discussed in this section. The parameter values used for the numerical simulations in Part 1 is provided in Table 2. Fig 2(A)-2(E) depict the nature of

**Table 2. Parameter values for numerical study.**

| Parameters | Value | Source |
|---|---|---|
| $a$ | 1191 | Estimated |
| $b$ | 0.98159 | [57] |
| $\alpha$ | 0.1 | Estimated |
| $\beta$ | 0.7 | [57] |
| $c$ | 0.0006 | [5] |
| $e$ | $0.1 \times 10^{-5}$ | [6] |
| $f$ | 0.0007 | [57] |
| $g$ | 0.05 | [57] |
| $h$ | 0.015 | Estimated |
| $k$ | 0.053 | [57] |
| $l$ | 0.012 | Model to fit |

all cases corresponding to $\phi = 0.98$. From the following figures, we have observed that the system is locally asymptotically stable at $E_0$.

## Part 2

To analyze the dynamical behavior of all people, the values of the parameters in Table 2 are employed. Fig 3(A)-3(F) depict all individuals' behavior over time for various fractional orders $\phi$. Fig 3(A) depicts that the number of susceptible individuals increases when $\phi$ changes from 0.8 to 0.95. An increase value of $\phi$ leads to decrease in the exposed rate in the exposed population in Fig 3(B). We see in Fig 3(C) that number of infected individuals increases when $\phi$ changes from 0.8 to 0.95. Fig 3(D) depicts that the number of quarantined individuals increases with time when $\phi$ decreases. The number of recovered individuals increases when $\phi$ changes from 0.8 to 0.95 in Fig 3(E). Fig 3(F) depicts that the number of death individuals increase with time when $\phi$ increases.

## Part 3

Part 3 of the numerical simulation investigates at how changing value impacts the fundamental reproduction number calculated in this work. Table 2 shows the parameter values utilized in the numerical simulations for Part 3. The acquired findings are shown in Table 3 after computing the fundamental reproduction numbers and utilizing the model parameters from Table 2. Table 3 shows that if a higher number of individuals in a community constantly utilize face masks, the COVID-19 epidemic can be decreased.

Fig 4 depicts that the values of $R_{Covid\ 19}$ decrease when $\alpha$ increase. The various consequences of wearing face masks were also investigated in this study, and it was discovered that wearing face masks on a consistent and suitable basis can inhibit the spreading of the COVID-19 pandemic.

The impact of $\alpha$ and $\phi$ on the Infected individuals ($I(t)$) is depicted in Fig 5(A) and 5(b). Based on the following figures, it can be noted that the implementation of maximum portion of population who use a face masks in order to effectively reduce COVID-19 transmission.

**Part 4.** This section describes the data matching and model validation of the system (3.3) for Infected instances. Table 2 depicts the parametric values.

Fig 6 depicts the graphical representation of the infected cases respectively of the model (3.3) and the real infected cases in Italy [see Table 4] from 1st January 2022 to 31st January 2022 [6].

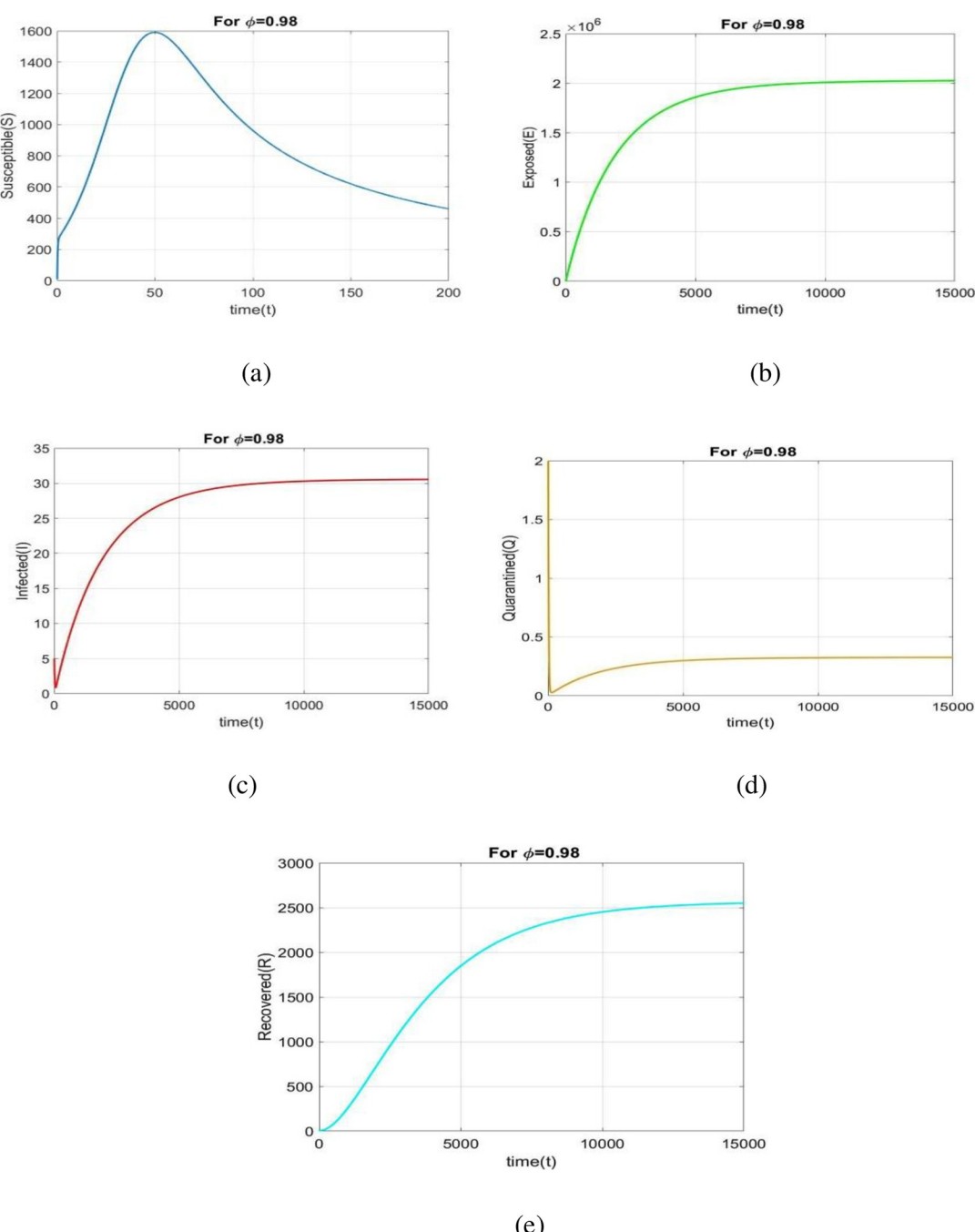

**Fig 2. Time series solution.** Time series of all classes correspondence to Table 2 taking $\phi$ = 0.98 of system (3.3).

## 7. Conclusion

The present study's possible goal is to develop a mathematical model for studying COVID-19 transmission patterns using actual pandemic cases in Italy, assisted by epidemiological modeling. The fractional order *SEIQRD* model was constructed and explored in this article in order to better explain the dynamics of the COVID-19 epidemic in Italy. We employed nonlinear analysis to demonstrate the model's existence and uniqueness. The model's fundamental

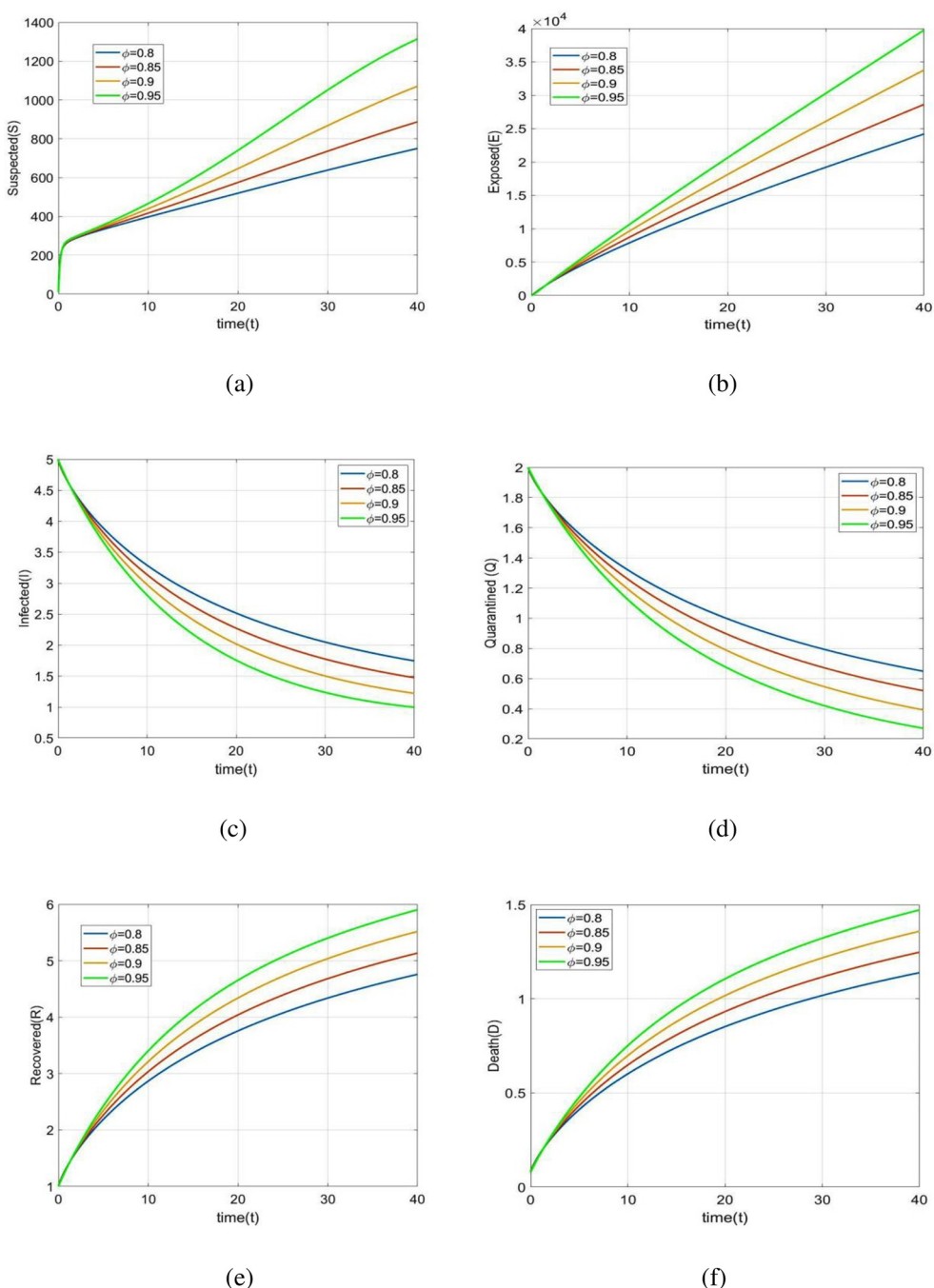

**Fig 3. Dynamical behavior.** Dynamic of all classes over time for various values of $\phi$ = 0.8, 0.85, 0.90, 0.95.

**Table 3. Numerical simulation of the varying effects of $\alpha$.**

| Parameter | Value | $R_{Covid\ 19}$ |
|---|---|---|
| $\alpha$ | 0.1 (10%) | 1.423 <1 |
| $\alpha$ | 0.5 (50%) | 0.973 <1 |
| $\alpha$ | 0.8 (80%) | 0.659 <1 |

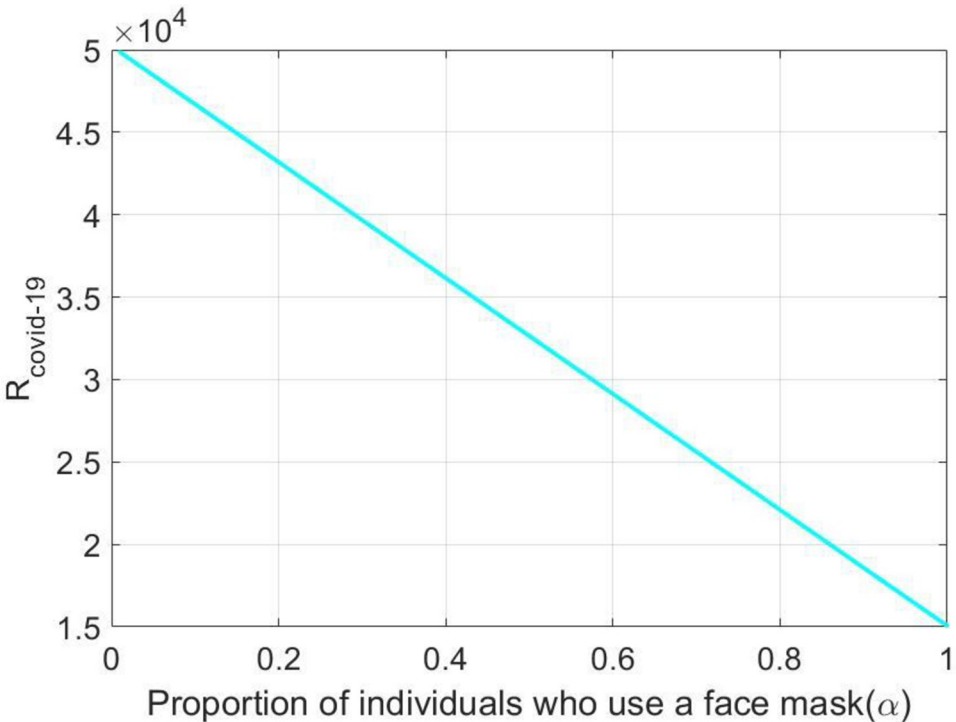

**Fig 4.** Variation of $R_{Covid\ 19}$ under $\alpha$.

reproduction number was also calculated using the next generation matrix technique. In order to stop the virus from spreading throughout the nation, our main goal is to establish the fundamental reproductive number and equilibrium. Furthermore, the global stability at the points $E_0$ and $E_1$ has been demonstrated. The results reveal that if $R_{Covid\ 19}<1$, the point $E_0$ is globally asymptotically stable. Also if $R_{Covid\ 19}>1$, the point $E_1$ is global asymptotic stable. Furthermore,

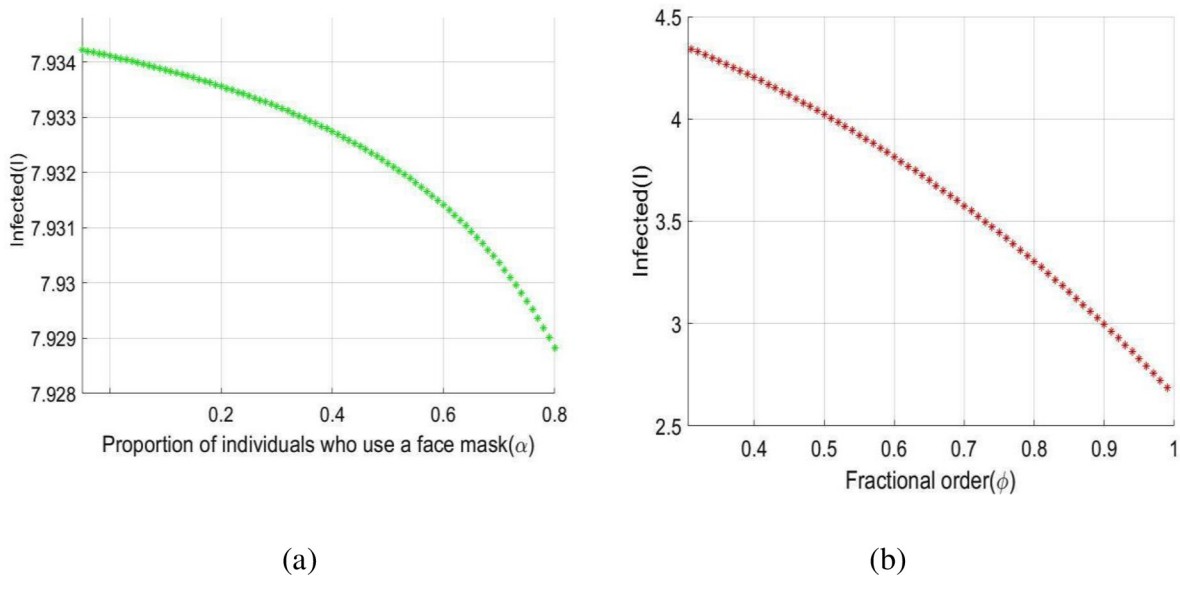

(a)

(b)

**Fig 5. Dynamics of $I(t)$ under $\alpha$ and $\phi$.**

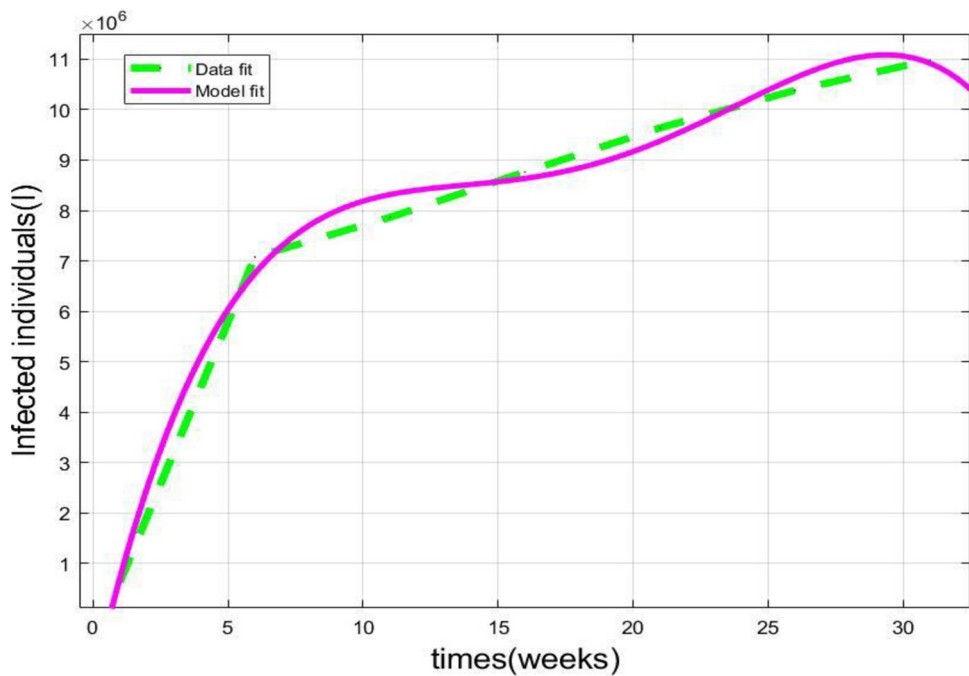

**Fig 6. Graph of the infected class of the proposed model (3.3) and real infected data [Table 4].**

using the fractional Taylor's approach, numerical analysis was done to establish an approximate solution for the suggested model. From the 1st of January 2022 to the 31st of January 2022, we have compared model values with real-world scenarios in Italy. Real data was also used to fit the model in order to forecast infected population instances in real life. In real-world dynamical processes, such as epidemic propagation, fractional calculus plays a vital role. The strength of memory effects, which is regulated by the order of fractional derivatives, is discovered to be dependent on the system dynamics. If the order of derivatives for the same set of parametric values is changed, the results will be changed (Fig 3). This study looked at the many consequences of wearing face masks, and it was discovered that wearing face masks on a consistent and suitable basis can help reduce the propagate of the COVID-19 disease (Figs 4 and 5). Currently, research on a vaccine to avert the COVID-19 pandemic is showing promising results, with Pfizer claiming that their vaccine has a 95% effectiveness rate. However, it will be some time before the vaccinations are widely distributed around the world. As a result, wearing a face mask should be made mandatory until everyone has access to vaccinations. Ministries and public health professionals may be able to develop strategic strategies to close

**Table 4. The number of Infected cases in Italy, from 1st January 2022 to 31st January 2022.**

| Day | Total reported data | Source |
| --- | --- | --- |
| 01/01/2022 | 635795 | [6] |
| 06/01/2022 | 7077458 | [6] |
| 11/01/2022 | 7864100 | [6] |
| 16/01/2022 | 8763280 | [6] |
| 21/01/2022 | 9637171 | [6] |
| 26/01/2022 | 10383904 | [6] |
| 31/01/2022 | 10983280 | [6] |

vaccination gaps and stop outbreaks in the future with the use of the research findings from the current study. Future studies should use the methodology provided in this work to the third wave of infected patients in Italy to assess the efficacy of existing COVID-19 prevention strategies.

## Acknowledgments

The authors are grateful to the reviewers for their valuable comments and suggestions.

## Author Contributions

**Conceptualization:** Banamali Roy.

**Data curation:** Animesh Mahata, Banamali Roy.

**Formal analysis:** Subrata Paul.

**Investigation:** Animesh Mahata.

**Methodology:** Subrata Paul, Animesh Mahata, Supriya Mukherjee, Prakash Chandra Mali.

**Software:** Animesh Mahata, Banamali Roy.

**Supervision:** Prakash Chandra Mali, Banamali Roy.

**Validation:** Animesh Mahata, Banamali Roy.

**Visualization:** Supriya Mukherjee.

**Writing – original draft:** Supriya Mukherjee.

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
