## [Decision Letter · Decision Letter 0]

6 Jul 2022

PONE-D-22-16861Fractional order  epidemic model of Covid-19: a case study of ItalyPLOS ONE

Dear Dr. Mahata,

Thank you for submitting your manuscript to PLOS ONE. After careful consideration, we feel that it has merit but does not fully meet PLOS ONE’s publication criteria as it currently stands. Therefore, we invite you to submit a revised version of the manuscript that addresses the points raised during the review process.

1) Authors should be prepared to incorporate major revisions.  

2) Authors should submit a list of responses to the comments and highlight them in revised manuscript.

3) Authors should avoid citing references that are not relevant to the research topic

We look forward to receiving your revised manuscript.

Kind regards,

Mohammed S. Abdo

Academic Editor

PLOS ONE

Journal Requirements:

6. Please ensure that you refer to Figure 1 in your text as, if accepted, production will need this reference to link the reader to the figure.

7. We note you have included a table to which you do not refer in the text of your manuscript. Please ensure that you refer to Tables 1 and 4 in your text; if accepted, production will need this reference to link the reader to the Table.

Additional Editor Comments:

Dear Authors,

we have received the reports from our advisors on your manuscript "Fractional order epidemic model of Covid-19: a case study of Italy"

(PONE-D-22-16861) which you submitted to "PLOS ONE".

Based on the advice received, I have decided that your manuscript could be reconsidered for publication should you be prepared to incorporate major revisions. When preparing your revised manuscript, you are asked to carefully consider the reviewers' comments which can be found in the system, and submit a list of responses to the comments and highlight them in revised manuscript.

Important note: The authors should not take into account references suggested by reviewers, except for only references that are closely related to the article.

Reviewers' comments:

Reviewer's Responses to Questions

**Comments to the Author**

1. Is the manuscript technically sound, and do the data support the conclusions?

Reviewer #1: Yes

Reviewer #2: Yes

Reviewer #3: Yes

2. Has the statistical analysis been performed appropriately and rigorously? 

Reviewer #1: Yes

Reviewer #2: Yes

Reviewer #3: Yes

3. Have the authors made all data underlying the findings in their manuscript fully available?

Reviewer #1: Yes

Reviewer #2: Yes

Reviewer #3: Yes

4. Is the manuscript presented in an intelligible fashion and written in standard English?

Reviewer #1: Yes

Reviewer #2: Yes

Reviewer #3: Yes

5. Review Comments to the Author

Reviewer #1: The presentation should be improved. Introduction need attention. Some recent work on COVID-19 relevant to this paper should be cited like: Chaos, Solitons & Fractals 157 (2022): 111955,Results in Physics 19 (2020): 103510.Alexandria Engineering Journal 59.5 (2020): 3221-3231.Results in physics 19 (2020): 103560.Results in Physics (2022): 105649

Reviewer #2: After reading this paper in detail, this paper needs to be revised via follows:

*The Abstract should contain answers to the following questions: What problem was studied and why is it important? What methods were used? What are the important results? What conclusions can be drawn from the results? What is the novelty of the work and where does it go beyond previous efforts in the literature?

*[12-16] these paper should be written in clearly

*"4.3 The Equilibrium Points of the system" this part of the paper needs to be extended a little more.

*They need to present more information about the novelty of this paper.

*What is the main contribution of this paper when we compare it via follows DOI:10.1080/17455030.2022.2075954; Modified Predictor–Corrector Method for the Numerical Solution of a FractionalOrder SIR Model with 2019-nCoV; Symmetry, 13(2428), 1-18, 2021;Chaos, Solitons and Fractals, 158(112050), 1-6, 2022; DOI:10.1142/S1793962323500083?

*What is the advantage of this operatır used in this paper?

*For this chapter of the paper "4.2 Non-negativity and boundedness of proposed model", they didnt cite any paper. Thus, they need to cite at least one paper related to this part of the paper.

After these modifications, this paper may be accepted.

Reviewer #3: Reviewer Report on:

Title: Fractional order SEIQRD epidemic model of Covid-19: a case study of Italy

Journal: PLOS ONE

Manuscript ID: PONE-D-22-16861

Authors: Mahata et al.

Overall Comments:

The authors have examined the fractional order SEIQRD compartmental model in a fractional order

framework to account for the uncertainty caused by the lack of information regarding the Coronavirus

(COVID-19). In addition, the fractional-order Taylor’s approach is utilized to approximate the solution

to the proposed model. The model’s validity is demonstrated by comparing real-world data with simulation outcomes. Furthermore, the system’s analysis and numerical findings show that regular usage of

face masks can help to reduce the COVID-19 epidemic.

However, it needs some revisions from the point of authors and readers to improve the quality of the

paper. After these MAJOR REVISIONS, I suggest that this paper can be accepted to publish in ”

PLOS ONE”.

My comments are as follows:

1. Ensure the end of each line of the equations has a punctuation, either a comma or a full stop if it is

the end of the equation - these are missing in several of the equations, see for example:

2. The Introduction should make a compelling case for why the study is useful along with a clear

statement of its novelty or originality by providing relevant information and providing answers to

basic questions such as:

• What is already known in the open literature?

• What needs to be done, why and how?

3. Clear statements of the novelty of the work should also appear briefly in the Abstract and Conclusions sections.

4. How did they decide the Lyapunov function in the global stability analysis?

5. How did they decide determining the initial conditions?

6. What about the parameter values in the system. Where they have been taken them from?

7. Whay did they ignore the fractional order in the system? Is it dimentionally correct now? Discuss

the system by comparing the following literature.

• Fractional order modelling of omicron SARS-CoV-2 variant containing heart attack effect

using real data from the United Kingdom. Chaos, Solitons & Fractals, 111954, (2022).

• Investigation of interactions between COVID-19 and diabetes with hereditary traits using real

data: A case study in Turkey. Computers in biology and medicine, 105044, (2021).

8. What is the main motivation in using the Caputo operator? What is the main advantages of it on

other existing operators with or without singular kernels?

9. The authors are requested to add more details regarding their original contributions in this manuscript.

10. Some figures should be redrawn clearly.

11. Authors should improve the introduction by including the recent development within the frame of

COVID-19 pandemic and its variants with the help of recently published papers. I recommend the

papers:

• Modeling and analysis of COVID-19 epidemics with treatment in fractional derivatives using real data

from Pakistan. The European Physical Journal Plus, 135(10), 1-42, (2020).

• Stability analysis of an incommensurate fractional-order SIR model. Mathematical Modelling and

Numerical Simulation with Applications, 1(1), 44-55, (2021).

• A new mathematical modeling of the COVID-19 pandemic including the vaccination campaign. Open

Journal of Modelling and Simulation, 9(3), 299-321, (2021).

• Vaccination effect conjoint to fraction of avoided contacts for a Sars-Cov-2 mathematical model.

Mathematical Modelling and Numerical Simulation with Applications, 1(2), 56-66, (2021).

• Extinction and stationary distribution of a stochastic COVID-19 epidemic model with time-delay.

Computers in Biology and Medicine, 105115, 141, (2022).

• Dynamics of cholera disease by using two recent fractional numerical methods. Mathematical Modelling and Numerical Simulation with Applications, 1(2), 102-111, (2021).

• Chaos of calcium diffusion in Parkinson’s infectious disease model and treatment mechanism via

Hilfer fractional derivative. Mathematical Modelling and Numerical Simulation with Applications,

1(2), 84-94, (2021).

• Fractional-order mathematical modelling of cancer cells-cancer stem cells-immune system interaction

with chemotherapy. Mathematical Modelling and Numerical Simulation with Applications, 1(2), 67-

83, (2021).

12. What is the novelty of your work? There are some similar papers which investigated in this area,

although there are some minor differences in the structure and applications to the COVID-19

models. Please state it clearly.

13. Have you employed any specific assumptions in your COVID-19 model? Please explain briefly.

14. More biological interpretations should be given. Please provide corresponding explanations of the

figures in terms of their biological meanings and the pointing out the novelty of the paper. How

do figures support your scheme? It will be more helpful to readers to have some discussions about

insight of the main results and outcomes of the figures.

Briefly, I recommend publishing after doing above MAJOR REVISIONS. So, I want to read speedily

the revised version of paper before publishing if it is possible for you.

With many thanks and best regards. . .

6. PLOS authors have the option to publish the peer review history of their article (what does this mean?). If published, this will include your full peer review and any attached files.

Reviewer #1: No

Reviewer #2: No

Reviewer #3: No

---

## [Author Response · Author response to Decision Letter 0]

16 Oct 2022

Manuscript ID: PONE-D-22-16861

Reply to the reviewers’ comments on the manuscript entitled “Fractional order SEIQRD epidemic model of Covid-19: a case study of Italy” submitted to the Journal “PLOS ONE”.

Reviewer #1: The presentation should be improved. Introduction need attention. Some recent work on COVID-19 relevant to this paper should be cited like: Chaos, Solitons& Fractals 157 (2022): 111955,Results in Physics 19 (2020): 103510.Alexandria Engineering Journal 59.5 (2020): 3221-3231.Results in physics 19 (2020): 103560.Results in Physics (2022): 105649

Authors’ response: Authors have modified the introduction part; some important recent and relevant references along with the one mentioned by the reviewer has been included.

Reviewer #2: After reading this paper in detail, this paper needs to be revised via follows:

*The Abstract should contain answers to the following questions: What problem was studied and why is it important? What methods were used? What are the important results? What conclusions can be drawn from the results? What is the novelty of the work and where does it go beyond previous efforts in the literature?

Authors’ response: Authors have modified the abstract as per the reviewer’s suggestions.

*[12-16] these papers should be written in clearly

Authors’ response: We have rewritten the description of these papers [12-16].

*"4.3 The Equilibrium Points of the system" this part of the paper needs to be extended a little more.

Authors’ response: Authors have modified this section (4.3 The Equilibrium Points of the system).

*They need to present more information about the novelty of this paper.

Authors’ response: Authors have modified the introduction part. The novelty and contribution of the work is clearly explained. 

*What is the main contribution of this paper when we compare it via follows DOI:10.1080/17455030.2022.2075954; Modified Predictor–Corrector Method for the Numerical Solution of a FractionalOrder SIR Model with 2019-nCoV; Symmetry, 13(2428), 1-18, 2021;Chaos, Solitons and Fractals, 158(112050), 1-6, 2022; DOI:10.1142/S1793962323500083?

Authors’ response: The authors would like to draw your attention towards the fact that the innovativeness of the paper lies in the consideration of the fractional order derivative of the population compartments. It is observed that fractional order derivatives reflect better results than integral order. Also the authors have modified abstract and introduction part; some important recent and relevant references along with the one mentioned by the reviewer has been included.

*What is the advantage of this operator used in this paper?

Authors’ response: Fractional derivatives are a powerful tool for describing memory and heredity characteristics in a wide range of systems and phenomena. Fractional-order differential equations store the function's comprehensive information in stacked form. Caputo fractional differentiations and differential equations have quite a number of useful advantages. When dealing with real-world problems, the Caputo derivative is particularly useful since it allows traditional initial and boundary conditions and the derivative of a constant is zero, which is not the case with the Riemann–Liouville fractional derivative.

*For this chapter of the paper "4.2 Non-negativity and boundedness of proposed model", they didnt cite any paper. Thus, they need to cite at least one paper related to this part of the paper.

Authors’ response: Authors have added one citation in this part.

Reviewer #3: Reviewer Report on:

Title: Fractional order SEIQRD epidemic model of Covid-19: a case study of Italy

Journal: PLOS ONE

Manuscript ID: PONE-D-22-16861

Authors: Mahata et al.

Overall Comments:

The authors have examined the fractional order SEIQRD compartmental model in a fractional order framework to account for the uncertainty caused by the lack of information regarding the Coronavirus (COVID-19). In addition, the fractional-order Taylor’s approach is utilized to approximate the solution to the proposed model. The model’s validity is demonstrated by comparing real-world data with simulation outcomes. Furthermore, the system’s analysis and numerical findings show that regular usage of face masks can help to reduce the COVID-19 epidemic.

However, it needs some revisions from the point of authors and readers to improve the quality of the paper. After these MAJOR REVISIONS, I suggest that this paper can be accepted to publish in ” PLOS ONE”.

My comments are as follows:

1. Ensure the end of each line of the equations has a punctuation, either a comma or a full stop if it is the end of the equation - these are missing in several of the equations, see for example:

Authors’ response: We are extremely sorry for that. All such errors have been edited and re-edited.

2. The Introduction should make a compelling case for why the study is useful along with a clear statement of its novelty or originality by providing relevant information and providing answers to basic questions such as:

• What is already known in the open literature?

• What needs to be done, why and how?

Authors’ response: The novelty and contribution of the work is clearly explained. The Introduction part of the manuscript contains extensive discussion of previous and related work along with relevant references. The research method adopted in this work are however original. 

3. Clear statements of the novelty of the work should also appear briefly in the Abstract and Conclusions sections.

Authors’ response: Authors have modified the introduction part. The novelty and contribution of the work is clearly explained.

4. How did they decide the Lyapunov function in the global stability analysis?

Authors’ response: An important technique in stability theory for differential equations is known as the direct method of Lyapunov. Let us consider a nonlinear time-invariant system, x' = f(x), where f(x) is assumed to be locally Lipschitz in x. Let xe be the equilibrium point of the system such that f(xe) = 0. The Lyapunov's Direct Method generally says that if you can find a continuous differentiable function V(x) satisfying the following conditions:

 V(x) > 0 (positive definite) and V(0) = 0,

 V'(x) = (∂V/∂x)•f(x) ≤ – W(x) ≤ 0,

 V(x) → ∞ as ||x|| → ∞.

Then the equilibrium point xe is globally stable if W(x) ≥ 0 (positive semi-definite), and globally asymptotically stable if W(x) > 0 (positive definite) for all x ≠ 0.

5. How did they decide determining the initial conditions?

Authors’ response: Authors have modified the numerical section.

6. What about the parameter values in the system. Where they have been taken them from?

Authors’ response: The references for parameters in Table 2 are reflected clearly in the last column of Table 2 in the numerical section.

7. Why did they ignore the fractional order in the system? Is it dimentionally correct now? Discuss the system by comparing the following literature.

• Fractional order modelling of omicron SARS-CoV-2 variant containing heart attack effect

using real data from the United Kingdom. Chaos, Solitons& Fractals, 111954, (2022).

• Investigation of interactions between COVID-19 and diabetes with hereditary traits using real data: A case study in Turkey. Computers in biology and medicine, 105044, (2021).

Authors’ response: We admit of a gross mistake in ignoring the parameters instead of redefining them. Necessary corrections have been made. We are extremely thankful to the reviewer for drawing our attention towards the relevant issue. 

8. What is the main motivation in using the Caputo operator? What are the main advantages of it on other existing operators with or without singular kernels?

Authors’ response: Fractional order modelling is a useful tool that has been used to explore the nature of diseases since the fractional derivative is an extension of the integer-order derivative. In order to replicate real-world issues, several innovative fractional operators with various properties have been designed. In addition, the integer derivative has a local identity, whereas the fractional derivative has a global character. The Caputo derivative is particularly useful for discussing real-world situations since it permits traditional beginning and boundary conditions to be used in the derivation, and the derivative of a constant is zero, whereas the Riemann–Liouville fractional derivative does not. Numerous varieties of fractional derivatives, both with and without singular kernels, are available today. Leibniz's query from 1695 marks the beginning of the fractional derivative. The fractional derivative also improves in the improvement of the system's consistency domain. We have the derivatives of Caputo, Riemann-Liouville, and Katugampola for singular kernels.There are two varieties of fractional derivatives without singular kernels: the Caputo-Fabrizio fractional derivative, which has an exponential kernel, and the Atangana-Baleanu fractional derivative, which has a Mittag-Leffler kernel. 

9. The authors are requested to add more details regarding their original contributions in this manuscript.

Authors’ response: The Introduction part of the manuscript contains extensive discussion of previous and related work along with relevant references. The research method adopted in this work are however original. 

10. Some figures should be redrawn clearly.

Authors’ response: Almost all figures have been well explained to get a better clarity regarding the biological aspect.

11. Authors should improve the introduction by including the recent development within the frame of COVID-19 pandemic and its variants with the help of recently published papers. I recommend the papers:

• Modeling and analysis of COVID-19 epidemics with treatment in fractional derivatives using real data from Pakistan. The European Physical Journal Plus, 135(10), 1-42, (2020).

• Stability analysis of an incommensurate fractional-order SIR model. Mathematical Modelling and Numerical Simulation with Applications, 1(1), 44-55, (2021).

• A new mathematical modeling of the COVID-19 pandemic including the vaccination campaign. Open Journal of Modelling and Simulation, 9(3), 299-321, (2021).

• Vaccination effect conjoint to fraction of avoided contacts for a Sars-Cov-2 mathematical model. Mathematical Modelling and Numerical Simulation with Applications, 1(2), 56-66, (2021).

• Extinction and stationary distribution of a stochastic COVID-19 epidemic model with time-delay. Computers in Biology and Medicine, 105115, 141, (2022).

• Dynamics of cholera disease by using two recent fractional numerical methods. Mathematical Modelling and Numerical Simulation with Applications, 1(2), 102-111, (2021).

• Chaos of calcium diffusion in Parkinson’s infectious disease model and treatment mechanism via Hilfer fractional derivative. Mathematical Modelling and Numerical Simulation with Applications, 1(2), 84-94, (2021).

• Fractional-order mathematical modelling of cancer cells-cancer stem cells-immune system interaction with chemotherapy. Mathematical Modelling and Numerical Simulation with Applications, 1(2), 67- 83, (2021).

Authors’ response: Authors have modified the introduction part; some important recent and relevant references along with the one mentioned by the reviewer has been included.

12. What is the novelty of your work? There are some similar papers which investigated in this area, although there are some minor differences in the structure and applications to the COVID-19 models. Please state it clearly.

Authors’ response: Authors have modified the introduction part. The novelty and contribution of the work is clearly explained.

13. Have you employed any specific assumptions in your COVID-19 model? Please explain briefly.

Authors’ response: The Introduction part of the manuscript contains extensive discussion of previous and related work along with relevant references. The research method adopted in this work are however original. 

14. More biological interpretations should be given. Please provide corresponding explanations of the figures in terms of their biological meanings and the pointing out the novelty of the paper. How do figures support your scheme? It will be more helpful to readers to have some discussions about insight of the main results and outcomes of the figures.

Briefly, I recommend publishing after doing above MAJOR REVISIONS. So, I want to read speedily the revised version of paper before publishing if it is possible for you.

Authors’ response: Figures have been well explained to get a better clarity regarding the biological aspect.

The authors are thankful to the reviewers and editor for their valuable comments and suggestions which also helped in improving the quality of our work.

---

## [Decision Letter · Decision Letter 1]

28 Nov 2022

Fractional order  epidemic model of Covid-19: a case study of Italy

PONE-D-22-16861R1

Dear Dr. Mahata,

We’re pleased to inform you that your manuscript has been judged scientifically suitable for publication and will be formally accepted for publication once it meets all outstanding technical requirements.

Kind regards,

Pablo Martin Rodriguez

Academic Editor

PLOS ONE

Reviewers' comments:

Reviewer's Responses to Questions

**Comments to the Author**

1. If the authors have adequately addressed your comments raised in a previous round of review and you feel that this manuscript is now acceptable for publication, you may indicate that here to bypass the “Comments to the Author” section, enter your conflict of interest statement in the “Confidential to Editor” section, and submit your "Accept" recommendation.

Reviewer #2: All comments have been addressed

Reviewer #3: All comments have been addressed

2. Is the manuscript technically sound, and do the data support the conclusions?

Reviewer #2: (No Response)

Reviewer #3: Yes

3. Has the statistical analysis been performed appropriately and rigorously? 

Reviewer #2: (No Response)

Reviewer #3: N/A

4. Have the authors made all data underlying the findings in their manuscript fully available?

Reviewer #2: (No Response)

Reviewer #3: Yes

5. Is the manuscript presented in an intelligible fashion and written in standard English?

Reviewer #2: (No Response)

Reviewer #3: Yes

6. Review Comments to the Author

Reviewer #2: (No Response)

Reviewer #3: After reading the revised form of the manuscript, I am convinced that the authors have improved their previous version that making it now suitable for acceptance by the PLOS ONE journal.

7. PLOS authors have the option to publish the peer review history of their article (what does this mean?). If published, this will include your full peer review and any attached files.

Reviewer #2: No

Reviewer #3: No

---

## [Editor Report · Acceptance letter]

1 Dec 2022

PONE-D-22-16861R1 

Fractional order *SEIQRD* epidemic model of Covid-19: a case study of Italy 

Dear Dr. Mahata:

I'm pleased to inform you that your manuscript has been deemed suitable for publication in PLOS ONE. Congratulations! Your manuscript is now with our production department. 

Kind regards, 

on behalf of

Professor Pablo Martin Rodriguez 

Academic Editor

PLOS ONE